# Accuracy and Acceptability of Wearable Motion Tracking for Inpatient Monitoring Using Smartwatches

**DOI:** 10.3390/s20247313

**Published:** 2020-12-19

**Authors:** Chaiyawan Auepanwiriyakul, Sigourney Waibel, Joanna Songa, Paul Bentley, A. Aldo Faisal

**Affiliations:** 1Brain & Behaviour Lab, Department of Computing, Imperial College London, London SW7 2AZ, UK; Chaiyawan.auepanwiriyakul16@imperial.ac.uk (C.A.); sigourney.waibel@imperial.ac.uk (S.W.); 2Behaviour Analytics Lab, Data Science Institute, London SW7 2AZ, UK; 3Department of Brain Sciences, Imperial College London, London W12 0NN, UK; Joanna.i.s@hotmail.co.uk; 4UKRI CDT in AI for Healthcare, Imperial College London, London SW7 2AZ, UK; 5MRC London Institute of Medical Sciences, London W12 0NN, UK

**Keywords:** healthcare, clinical monitoring, hospital, stroke, human activity recognition, natural movements, inertial movement unit, optical motion tracking

## Abstract

Inertial Measurement Units (IMUs) within an everyday consumer smartwatch offer a convenient and low-cost method to monitor the natural behaviour of hospital patients. However, their accuracy at quantifying limb motion, and clinical acceptability, have not yet been demonstrated. To this end we conducted a two-stage study: First, we compared the inertial accuracy of wrist-worn IMUs, both research-grade (Xsens MTw Awinda, and Axivity AX3) and consumer-grade (Apple Watch Series 3 and 5), and optical motion tracking (OptiTrack). Given the moderate to strong performance of the consumer-grade sensors, we then evaluated this sensor and surveyed the experiences and attitudes of hospital patients (N = 44) and staff (N = 15) following a clinical test in which patients wore smartwatches for 1.5–24 h in the second study. Results indicate that for acceleration, Xsens is more accurate than the Apple Series 5 and 3 smartwatches and Axivity AX3 (RMSE 1.66 ± 0.12 m·s^−2^; R^2^ 0.78 ± 0.02; RMSE 2.29 ± 0.09 m·s^−2^; R^2^ 0.56 ± 0.01; RMSE 2.14 ± 0.09 m·s^−2^; R^2^ 0.49 ± 0.02; RMSE 4.12 ± 0.18 m·s^−2^; R^2^ 0.34 ± 0.01 respectively). For angular velocity, Series 5 and 3 smartwatches achieved similar performances against Xsens with RMSE 0.22 ± 0.02 rad·s^−1^; R^2^ 0.99 ± 0.00; and RMSE 0.18 ± 0.01 rad·s^−1^; R^2^ 1.00± SE 0.00, respectively. Surveys indicated that in-patients and healthcare professionals strongly agreed that wearable motion sensors are easy to use, comfortable, unobtrusive, suitable for long-term use, and do not cause anxiety or limit daily activities. Our results suggest that consumer smartwatches achieved moderate to strong levels of accuracy compared to laboratory gold-standard and are acceptable for pervasive monitoring of motion/behaviour within hospital settings.

## 1. Introduction

Wearable movement sensors have the potential to transform how we measure clinical status and wellbeing in everyday healthcare. Tracking patient movements can help characterise, quantify, and monitor physical disability; highlight deteriorations; and signal treatment response. Remote patient assessment may also allow for more cost-effective monitoring and offer advantages in contexts where direct contact is restricted e.g., due to Covid-19 associated isolation. Currently, behavioural assessments in clinical settings are characterised by intermittent, time-consuming human observations, using inconsistent subjective descriptions [1]. With an ageing population and increasing health system costs, there is a growing interest in seeking low-cost, automated methods for observing and quantifying patient behaviours [2,3,4,5,6,7]. Presently, the two leading wearable sensors offered for automated motion tracking are (1) camera-based optical tracking systems and (2) body-worn Inertial Measurement Units (IMUs), consisting of a triaxial accelerometer, a triaxial gyroscope, and, frequently, a magnetometer that record linear accelerations, angular velocities and magnetic field strength in a three-dimensional (3D) Cartesian space.

Body-worn IMUs hold several advantages over optical systems for behaviour tracking ‘in the wild’ (i.e., free-living conditions). While optical systems are considered the gold-standard for spatial movement tracking in controlled laboratory environments, the restriction on cameras’ field of views, obscuration of reflective markers, and lighting confounds introduce noises that compound when estimating the rigid body orientation [4]. Furthermore, optical equipment is expensive, cumbersome, complex to calibrate and operate, and has limited usage duration [4]. These limitations render optical motion tracking systems impractical for clinical use. In contrast, IMU sensors offer low cost, highly portable, robust and inconspicuous alternatives that are better suited to measure daily life activities in unconstrained environments such as hospitals and care homes [8,9,10]. A diverse range of wearable IMUs is commercially available, which can be broadly grouped into consumer-grade products such as wrist-worn fitness trackers or smartwatches and research-grade IMU sensors for research or clinical purposes [11]. Consequently, widely adopted commercial products with networking functionalities are increasingly being applied for motion tracking applications with the advantages of being ubiquitous, relatively low-cost, robust, easily cleanable and simple to self-apply and operate [11,12]. Both fitness bands and smartwatches fall within this wearable category. We focus here on smartwatches as they are more easily programmable and facilitate distribution and update of custom software through App stores, thus making them attractive as a platform for wearable research and development. Smartwatches are already increasingly employed for health monitoring purposes, and so there is a growing need to assess their measurement precision against gold-standard references.

Presently, the use of consumer smartwatches in health applications is limited by the unknown data quality of their IMU data and their evaluation in a research or clinical setting. Previous work [13,14,15,16,17,18,19] focused on validating built-in heart rate, energy expenditure, and step count measurements relative against ground-truth measurements of electrocardiography, indirect calorimetry and observed step counts. For instance, Wallen and colleagues [13] found that the smartwatches, Apple Watch (Apple Inc., Cupertino, CA, USA), Fitbit Charge (Fitbit Inc., San Francisco, CA, USA), Samsung Gear S (Samsung, Seoul, Korea) and Mio Alpha (MioLabs Inc., Santa Clara, CA, USA) underestimated outcome measurements such as step count in terms of average error range between 4% to 7% (Apple Watch error = −4.82%, Fitbit Charge HR error = −5.56%, and Samsung Gear S error = −7.31% (relative errors computed from the raw data provided in the paper).

In evaluating the accuracy and precision of smartwatch IMUs, however, both absolute errors (i.e., how much the sensor differed from the ground-truth value) and correlations (i.e., how well does the sensor track the dynamic changes of ground-truth values) need to be measured. Apple Watch (r = 0.70), Fitbit Charge HR (r = 0.67) and Samsung Gear S (r = 0.88) were shown to correlate reasonably well with ground-truth step count [13]. Note, out of the three smartwatches, the one with the highest average error also shows the best performance in tracking step count dynamically, so signal quality rankings for the same watch differs across accuracy and precision measures. Moreover, across sensing modalities that were not related to kinematics, the same set of watches performed differently in how well they captured heart rate and energy expenditure, and no single smartwatch was the best in overall assessed modalities.

Validating smartwatch-derived measures for clinical or scientific use is complicated as most measured outcomes recorded from consumer wearables e.g., built-in energy expenditure and step counts are derived from undisclosed, proprietary algorithms with unknown modelling assumptions that have not gone through medical certification processes. Assuming that different generations of the same smartwatch models are not significantly different from each other across studies, it suggests that experimental protocols play a considerable impact in assessing measurement quality. Therefore, the assessment and direct comparison of devices require a defined and reproducible description of the assessment process. Another study [18], compared fitness armbands and smartwatches to the gold-standard measurements, found that even average step count errors varied widely between wearable devices, such as Nike+ FuelBand (Nike Inc., Beaverton, Oregan) (error = 18.0%), Jawbone UP (Jawbone, San Francisco, CA, USA) (error = 1.0%), Fitbit Flex (Fitbit Inc., San Francisco, CA, USA) (error = 5.7%), Fitbit Zip (Fitbit Inc., San Francisco, CA, USA) (error = 0.3%), Yamax Digiwalker SW-200 (Yamasa Tokei Keiki Co. Ltd., Tokyo, Japan) (error = 1.2%), Misfit Shine (Misfit Wearables Inc., San Francisco, CA, USA) (error = 0.2%), and Withings Pulse (Withings, Issy-les-Moulineaux, France) (error = 0.5%). This shows that even wearable sensors developed specifically for tracking fitness activities such as walking and running activity made some errors, although it is unclear if a more generous counting of steps may be of fitness or commercial interest as it depends on the context. For instance, a clinical system designed to identify functional deterioration might prefer underestimation as it minimises false-positive errors; whereas a consumer fitness tracker might favour overestimation as it overinflates user successes. The degree of error permissible will also differ between whether a method is intended for a professional-clinical setting or a consumer-lifestyle.

The lingering uncertainty in the quality and usability of the underlying sensor signal quality motivated our work here, by first evaluating the raw IMU sensor data quality of smartwatches, and then second, trial the feasibility of large-scale deployment in a clinical care setting through the patient (PPI) and healthcare worker involvement. Clinical and care wearable applications and analysis rely upon quantified, regulatory acceptable measures of accuracy of the fundamental signal (linear acceleration for accelerometers and angular velocity for gyroscopes) that must be compared to the reference standards. The quality in these fundamental signals allows us to assess how well they can in principle track measures, such as body kinematics, but also more indirectly inferred measures, often clinical outcome measures and primary endpoints of clinical trials (such as step counts). To date, there has been no independent direct comparison between common consumer smartwatches, research-grade IMUs inertial and ground truth optical motion tracking. This is in part due to consumer smartwatch closed-system barriers to raw IMU data extraction, which we overcome through developing customised software. We also developed an easily reproducible measurement protocol to directly assess and compare smartwatches in naturalistic movement tasks, performed by the same human on all compared devices at the same time. Additionally, a separate issue for the clinical feasibility of smartwatches in care is the feasibility of their deployment and their practicality and acceptability in everyday use by both patients and healthcare workers. Research to date on patient and healthcare staff attitudes towards the continuous wearing of IMU sensors is scarce. While some studies report user-perceptions (e.g., user-friendliness and satisfaction) of smartwatches and fitness devices [20,21,22,23,24,25], these often focus on community settings, chronic disease, young/middle-aged subjects, and healthy participants and, as such, are not as relevant for typical in-patient populations.

We focused among the consumer-grade smartwatches on a single smartwatch make, so we could evaluate the technology within a large, parallel deployment of units in the care setting while remaining within a reasonable budget. We used market share as a guide for deciding which smartwatch to evaluate: Apple Watch holds the most share in the market (47.9% market share), while the second most popular device, Samsung Gear, retains only 13.4% of the global market share (in terms of shipment units) in the first quarter of 2019 [26].

## 2. Materials and Methods

### 2.1. Material

Both the sensor signal quality comparison and the sensor acceptability study we explored here were based on the Apple Watch (Series 3 and 5), which is the leader in consumer smartwatches in terms of market share in 2019/2020 (47.9%) [26]. We compared the inertial accuracy of the consumer smartwatch IMUs (from Apple Watch Series 3 and 5) against two well-known research- and clinical-grade IMU sensors: 1. Xsens MTw Awinda (Xsens Technology B.V., Enschede, The Netherlands) with many published applications and validation studies in biomechanics (e.g., [27,28,29,30,31,32,33]) and 2. Axivity AX3 (Axivity Ltd., Newcastle Upon Tyne, UK) [34,35] with many published biomedical research applications including its deployment in the UK Biobank cohort with over 3500 devices used by 100,000 participants (e.g., [36]). Additionally, we also compare the IMUs against a gold-standard for human movement assessment in the form of optical motion tracking (OptiTrack, Natural Point Inc., Corvallis, OR, USA) [37,38,39].

### 2.2. Sensor Signal Quality Study

#### 2.2.1. Population

We recruited a sample of healthy volunteers (*n* = 15) from Imperial College London to take part in a sensor accuracy assessment study. All participants agreed to take part with no withdrawals. All participants gave informed consent to participate in the study, and the study was ethically approved by the Imperial College London University Science, Engineering and Technology Research Ethics Committee (ICREC).

#### 2.2.2. Data Collection

To record and extract inertial data from the Apple Watches (Series 3 and 5), we developed a piece of software, a WatchOS App (See Appendix B
*WatchOS Application and Server*, for implementation detail), to collect real-time triaxial acceleration (±8 g for Series 3 and ±16 g for Series 5) and triaxial angular velocity data (±1000 degree/s for Series 3 and ±2000 degree/s for Series 5) at 100 Hz. The watch stored data to an onboard memory and offloaded the data to a custom-configured base station wireless access point and laptop. The Xsens MTw Awinda unit recorded packet-stamped triaxial acceleration (±16 g), triaxial angular velocity (±2000 degree/s), and triaxial magnetic fields (±1.9 Gauss) data at 100 Hz. The Xsens sensors wirelessly transmitted data in real-time to a base station and laptop which recorded the data within the MT Manager software. The Axivity AX3 unit recorded triaxial acceleration data at 100 Hz with a configurable range of ±2/4/8/16 g (±16 g was selected for this study). The Axivity sensors stored data to an onboard memory and offloaded the data upon attachment to a laptop via the installed AX3 OMGUI software. For the ground-truth optical motion tracking, we calibrated 4 ceiling-mounted OptiTrack cameras according to the manufacturer’s specifications and created a rigid body model using four reflective markers attached to the single marker pad (see photos in Figure 1a). The system wirelessly recorded both triaxial rotation of the constructed rigid body and absolute triaxial position of the markers and the rigid body within a 6 cubic meters area at 240 Hz and transferred the recorded data in real-time to a laptop running an optical motion capture software, Motive.

To construct the sensor stack, we used Apple Watch Series 3 as a base. We carefully aligned the centre of gravity of the OptiTrack markers pad to the centre of gravity of the base before affixing the pad to the base via a strong double-sided tape. We also utilised strong electrical tapes to enhance structural integrity by wrapping the resulting sensor stack tightly together. We then repeated the process for each sensor, each time utilising the previous stack as the base and carefully aligning their centre of gravity together (see Figure 1).

We asked each participant to perform a predefined sequence of upper body movements in a 6-min controlled exercise while data were simultaneously recorded from the sensor and marker stack illustrated in Figure 1. We chose the movement tasks outlined in Table A1 because they spanned the full range of natural joint angles at shoulder and elbow during a complex 2-joint movement typical for natural activities e.g., reaching for, passing, and picking up an object. Each participant trial consisted of a sequence of 4 distinct movements in time with a 120 BPM metronome. We constrained movement tasks within the OptiTrack cameras’ field of views (see Figure 2).

#### 2.2.3. Data Processing

Following the completion of the movement protocol, we collected and analysed sensor inertial data in MATLAB^®^ (MathWorks, Inc., Natick, MA, USA). All sensor data was linearly resampled to a constant sampling rate of 100 Hz. We manually inspected the OptiTrack data within Motive, corrected mislabelled markers, and reconstructed the rigid body data using the newly corrected makers. We derived OptiTrack triaxial accelerations from the positional data. We unrolled Xsens packet stamps and replaced missing packet rows with a Not-a-Number (NaN) row vector and calculated Xsens timestamps using the unrolled packet stamps.

A 0.25 to 2.5 Hz 6th-order Butterworth bandpass filter decontaminated signals data from integration drifts and differentiation noise. We determined the cut-off frequencies by plotting the power spectrums of acceleration (a) and position (b) signals of the sensors, which demonstrated that acceleration concentrate around 2 Hz and position around 0.5 and 1 Hz (Figure 3). This frequency is consistent with our movement tasks which were restricted to a lower bound of 0.5 Hz or an upper bound of 2 Hz.

We visually segmented sensor recordings using an identifiable movement event (five handclaps) at the start and end of each participant recording and further segmented each participant recording into the 4 movement tasks. To align signals spatially, we converted acceleration data from unit of gravity (g) to meters per second squared (m·s^−2^) using a constant of 9.80665 m·s^−2^. The angular velocity unit is in radians per second (rad·s^−1^). We used cross-correlation to calculate any lag between signals in each unique device pair to align signals temporally at the common starting point. To verify alignment, we applied cross-correlation to each signal pair. We then converted aligned signals into a 1D vector using a Euclidean norm function and removed NaN rows from each sensor vector and its corresponding sensor pair.

#### 2.2.4. Analysis

We compared vectors for each inertial sensor against every other inertial sensor and OptiTrack using Root-Mean-Square-Error (RMSE) and R-Squared (R^2^) metrics for acceleration and angular velocity. Axivity AX3 does not contain a gyroscope and we found that optical motion tracking angular velocity estimates were unreliable due to compounded noise during rigid body rotation estimations and so we omitted angular velocity comparisons for Axivity and optical motion tracking systems. We reported the results between each sensor pair in the format of Mean of Metrics across all trials ± Standard Error of Metrics across all trials. To assess whether the stacking of the sensors and markers introduced any recording error due to varied distance between the wrist and the sensor, we plotted the difference between acceleration and angular velocity from an Apple Watch (Series 3) stacked at the bottom and an Apple Watch (Series 5) stacked at the top of the sensor and marker stack (See Figure A1 for more information). We interpreted the strength of the R^2^ between sensors using the following descriptive categories: weak (R^2^ = < 0.5), moderate (R^2^ = 0.5–0.7) and strong (R^2^ ≥ 0.7) agreement [13].

### 2.3. Sensor Acceptability Study

#### 2.3.1. Population

We recruited a consecutive convenience sample of 44 in-patients and 15 healthcare professionals to complete patient and healthcare professional sensor acceptability questionnaires, respectively. We recruited patients who were enrolled in a study exploring relationships between free-living wearable motion sensor data and observed behaviour and clinical scores (data to be published separately). Inclusion criteria were: (1) admission to Stroke or Neurology Wards at Charing Cross NHS Hospital, London; (2) >18 years old; (3) absence of significant cognitive impairment; (4) Glasgow Coma Score >13; and (5) ability to understand English; (6) and capacity to consent. Exclusion criteria were: (1) inclusion within another interfering trial; and (2) contagious skin infections or other rash interfering with sensor placement. The study screened 71 potentially eligible patients, of which the following were subsequent reasons for further exclusion: swollen ankles (*n* = 2); same-day discharge (*n* = 5); and patient refusal to take part because they were not interested (*n* = 7) or felt fatigued or unwell (*n* = 8). A total of 49 patients completed the study protocol, of which 5 were excluded due to: a clinical need to remove watches early to undertake magnetic resonance imaging (MRI) (*n* = 2); and hospital discharge or transfer (*n* = 3). All participants gave informed consent to participate in the study and the study was ethically approved by the UK’s Health Research Authority IRAS Project ID: 78462.

#### 2.3.2. Data Collection

All healthcare professional and patient interviews took place in the ward by the patient bedside. The clinical researchers explained the purpose of the study both verbally and via written information sheets. Questionnaires lasted 10-min and two clinical researchers recorded the participants’ answers.

All in-patients wore 4 consumer smartwatches (Apple Watch Series 3) while they performed their usual everyday activities on the hospital ward as depicted in Figure 4. The IMU sensors were affixed to the patients at 4 locations: 1. outer right ankle 2. outer left ankle 3. upper right wrist and 4. upper left wrist. The sensors at the ankle location were placed just above the lateral malleolus. The ones on the wrist were located near the end of Ulna and Radius. We chose this arrangement to capture asymmetrical patterns of upper and lower limb weakness (that are typical for stroke); and to support recognition of different locomotions (e.g., walking, standing and sitting) and different daily activities entailing manual interactions (e.g., drinking and eating). To assess sensor wear-time acceptability over day-long continuous recording periods, we asked all participants to wear the sensors for a full working day and a random subset of the total (*n* = 11) to continue to wear the sensors overnight. Subject wear times varied (1.5–24 h), as watches were removed for patient showers, medical scans, and upon patient hospital discharge or transfers. At the end of the recording protocol, we asked participants to complete a questionnaire to collate their views regarding wearable technology. We derived and adapted [23,40,41] and agreed upon the final study questions via discussions between the clinical researchers, a Consultant Neurologist, a Stroke Physician and two patients. Watches were attached using a soft, breathable nylon replacement sport strap with an adjustable fastener. During the protocol, we locked watch user-functionalities, blanked out and covered the watch screen with a plastic sleeve to prevent user interaction. After sensor recordings, in-patients answered 10 close-ended questions outlined in Table 1.

We provided healthcare professionals with the intended functionality of the sensors for monitoring patient movement in the hospital and showed healthcare professionals how to operate the devices. Healthcare professionals then answered 5 closed-ended questions outlined in Table 1. Thereafter, we asked both in-patients and healthcare professionals open-ended questions described in Table 1.

Rating scales and descriptive category groupings were as follows:

icQ1–10 used a 1 to 7 rating scale: 1 to 2 (strongly disagree); 3 to 4 (somewhat agree); and 5 to 7 (strongly agree).hcQ1–2 used a 1 to 7 rating scale: 1 to 2 (strongly disagree); 3 to 4 (somewhat agree); and 5 to 7 (strongly agree).hcQ3 used a 1 to 10 rating scale: 0 to 5 (no opportunity); and 6 to 10 (great opportunity).hcQ4 used a 1 to 10 rating scale: 0 to 5 (no danger/safe); and 6 to 10 (danger).hcQ5 was collected with a −3 to +3 rating scale: −3 (would not use the intervention); −2 to 0 (would only use the intervention if controlled by a human caregiver); and 1 to 3 (would use the intervention and it could replace some interventions currently implemented by human caregivers).

#### 2.3.3. Analysis

We aggregated in-patient and healthcare professional responses for close-ended questions into the broader rating scale descriptive categories (e.g., strongly disagree, somewhat agree, strongly agree for the 1–7 rating scale) and calculated (1) the percentage of responses for each descriptive category for in-patients; and (2) the frequency of responses for each descriptive category for healthcare professionals separately. We assessed all open-ended responses by a thematic analysis which aimed to describe concepts extracted from the participant responses. Literal comments were (1) recorded by the two clinical researchers (2) compared and grouped based on similarity and creation and (3) subsequently merged into core agreed to themes via consensus between the researchers.

## 3. Results

We determined how all three IMUs compared to each other and to a gold-standard for human movement assessment in the form of optical motion tracking. The findings from the first study led to the selection of the smartwatch sensors for the second study in which we conducted a survey, assessing the experiences and attitudes of 44 in-patients and 15 healthcare professionals after a trial of continuous smartwatch use in hospital patients. Together these questions establish the scientific and practical validity of wearable inertial sensors for movement tracking in clinical applications, particularly within hospitals.

### 3.1. Sensor Signal Quality Study Results

A total of 15 healthy subjects (2 females; 13 males) were recruited from the students from within the Department of Bioengineering and the Department of Medicine, Imperial College London. Two subjects were excluded due to recording equipment malfunctions; one was excluded due to a missing video. We compared the inertial accuracy of both research-grade and consumer-grade IMUs relative to gold-standard optical motion tracking. The RMSEs for acceleration against OptiTrack ranged from 1.66 to 4.12 m·s^−2^. When comparing to OptiTrack acceleration, R^2^ agreement was stronger for Xsens than for Axivity, the latter of which had weak R^2^ agreement. Apple Watches only demonstrated weak to moderate R^2^ agreement with OptiTrack acceleration. When comparing to Xsens (research inertial sensor reference) acceleration, Apple watch demonstrated stronger R^2^ agreement than Axivity. When comparing to Xsens (research inertial sensor reference) angular velocity, Apple watch (Series 3 and 5) had strong R^2^ agreement. Apple Watch Series 3 and Apple Watch Series 5 had a strong agreement with each other for acceleration and angular velocity: The actual R2 for angular velocity comparisons (Figure 5c) between Apple Watch Series 3, Series 5, and Xsens MTw Awinda (which in the graphic are rounded to two digits) are 0.9911 ± 0.0006, 0.9957 ± 0.0003, and 0.9938 ± 0.0005, respectively, these are not visible in the figures due to rounding. See Figure A2, Figure A3, Figure A4 and Figure A5 for the result breakdown for each of the 4 movement tasks.

### 3.2. Sensor Acceptability Study Results

A total of 44 patients (50% female; average age: 64 years; interquartile age range: 24–92 years) completed the acceptability questionnaire. A further 15 healthcare professionals (66% female; doctors = 5, nurses = 4, therapists = 3, and healthcare assistants = 3) working directly with these patients were also recruited. Further details of in-patient and healthcare characteristics are outlined in Table A2 and Table A3 respectively.

The patient survey responses showed that in-patients in both age groups strongly agreed with all 10 closed-ended questions (icQ1–10), suggesting that sensors were easy to operate and learn to use, comfortable, did not limit daily activities, did not cause anxiety and unobtrusive in appearance as seen in Table 2. As illustrated below in Table 2, the survey of healthcare professionals showed high levels of agreement with statements that the system was easy to operate and learn to use (hcQ1–2) and presented no danger (hcQ4). Healthcare professionals were more split in their views regarding the opportunity of wearable tracking sensors and AI in healthcare delivery (hcQ3) and whether the technology could be used without the control of a human caregiver (hcQ5). A difference in opinions existed across the varied healthcare professional specialties. In particular, given strong evidence that an intervention was better or equivalent to current observations, some therapists (*n* = 2), nurses (*n* = 3), and healthcare assistants (*n* = 1) still viewed human control as important, whereas all doctors (*n* = 5) viewed human control as unnecessary. Additionally, all therapists (*n* = 3) viewed the increasing use of wearable motion sensors and artificial intelligence technologies as an opportunity for healthcare applications, whereas some doctors (*n* = 3), nurses (*n* = 2), and healthcare assistants (*n* = 1) thought it presented no opportunity. A significant proportion of in-patients reported that they felt neutral towards the sensors or had nothing in particular to comment when asked open questions about system likes (*n* = 18), dislikes (*n* = 34), and expected functions and characteristics (*n* = 31) (ioQ1–3). The comments from in-patients who did provide detailed responses were grouped into various themes (5 likes, 5 dislikes, 6 expected characteristics and functions) as outlined in Table 3.

In response to the open-ended question (hoQ1–4), healthcare professionals (*n* = 10) highlighted that the sensors would only cause discomfort to a selection of patients in certain situations (e.g., some cases of hemiparesis, swelling or long wear periods). All healthcare professionals viewed the system as not intrusive to healthcare professionals and, similarly, the majority of healthcare professionals (*n* = 12) also perceived that the sensors were not intrusive to patients. Six healthcare professionals commented that the system may interfere with medical treatments, while eight disagreed and thought it would not interrupt care needs. The open-ended comments from healthcare professionals were grouped into themes (7 benefits, 5 risks, 2 likes, 2 dislikes, 4 expected characteristics and functions) as outlined in Table 4.

## 4. Discussion

Wearable inertial sensors are increasingly exploited for clinical purposes by providing low-cost, pervasive, high-resolution tracking of natural human behaviour. However, their validity assumes that they convey accurate motion information, while their clinical feasibility and adoption require a minimal level of user acceptance among patients and healthcare professionals. In this study, we tested these two assumptions by: (1) quantifying the accuracy of commonly used wearable inertial sensors relative to each other and a gold-standard optical motion tracking instrument and; (2) surveying the attitudes of target healthcare professionals and in-patient populations following a trial period of continuous wearable inertial sensor use. As our study approached two original research questions, our results are not directly comparable to results of earlier literature as we addressed different research problems (raw sensor data quality) and employed standardised and consistent methods for collecting comparative data across devices. For example, we differed in our choice of outcome measure (we used straightforward inertial estimates rather than combined estimates e.g., [27,30]) and type of sensor (we used individual sensors rather than full-body sensor suits [29,32]). This is significant given that many [42] models depend upon using good raw accelerometer and gyroscope data.

### 4.1. Sensor Signal Quality Study

Relative to ground-truth optical motion tracking, the consumer smartwatches (i.e., the Apple Watch Series 3 and 5) and the research-grade IMU Xsens achieved cleaner linear acceleration signals and lower errors than Axivity (Figure 5). This is likely to be due to accelerometer and gyroscope fusion in the cases of Xsens and consumer smartwatches that enables superior isolation of gravity vectors from acceleration signals; whereas Axivity acts as a pure acceleration logger that relies on a low-pass filter to accomplish this [4,43,44]. We found that the consumer smartwatches had strong angular velocity agreement when compared against research inertial sensor reference (Figure 5). However, Xsens had stronger fidelity for recording accelerations (R^2^ = 0.78), perhaps due to the additional magnetometer and strap down integration (SDI) technology [45,46,47,48]. Accelerations and angular velocities were very similar between Apple Watch Series 3 and Series 5 (Figure 5), this provides a measure of confidence to pool and compare studies using IMU data recorded from different versions of the smartwatch.

The sensor signal quality experiment had several strengths. In contrast to earlier studies validating consumer sensor proprietary ‘black-box’ energy expenditure, heart rate, step count measures, and joint estimates, our study was unique in measuring the accuracy of the straightforward inertial movement measurements (i.e., acceleration and angular velocity) of the sensors. We were able to do this by developing custom software to bypass the consumer smartwatch closed system barriers to export raw acceleration and angular velocity data. Using our custom extraction and transmission of the smartwatch IMU data, the data could easily be integrated with wider systems for unique research and clinical applications outside of the laboratory. Studies [49,50,51] also developed custom software to export Apple Watch data but did not assess the IMU accuracy. Moreover, our validation of four individual Xsens MTw sensors accuracy, as opposed to the 17-sensor Xsens MVN BIOMECH full-body suit in earlier studies, is also noteworthy. We posit that the full body suit is less practical for long-term continuous ‘in the wild’ behaviour monitoring of in-patients as it is higher cost, more challenging to operate and calibrate, obtrusive for the user for long wear times (requires 17 sensors attached to various positions on the body), and requires mobile participants for the walking calibration. We evaluated whether the position of the sensor in the sensor stack (i.e., distance to the wrist) affected the captured movement estimates and found that there were minimal signal differences between the Apple watch at the top and bottom of the sensor stack, especially for the slower movement tasks.

Limitations of our sensor accuracy study include the fact that the movement task duration (~5 min) may not have sufficiently captured drift over longer time-periods and the controlled task of the upper limb may not have fully represented the range of natural human movements. Importantly, we note that some differences between optical motion tracking and Xsens could be explained by greater inertial sensor jitter and latency and horizontal position (XY) drift during stationary periods or exaggerated noise from marker obstruction through the differentiation when deriving acceleration from optical motion tracking position data [52].

In summary, our sensor quality study results demonstrate that: 1. sensors with both accelerometers and gyroscopes (Xsens and Apple Watches) perform better than just accelerometers (Axivity); 2. accuracy for acceleration is higher for research-grade sensors that employ SDI technology with magnetometry (Xsens) compared to non-SDI non-magnetometer sensors (Apple Watches); 3. Apple Watch Series 3 and Series 5 demonstrated high signal similarities suggesting cross-generation compatibility between Apple Watches inertial data. This gave us confidence that, with relatively few drawbacks, consumer-grade smartwatches can be objectively used within a clinical- and research-grade setting.

### 4.2. Sensor Acceptability Study

Similarly, our acceptability results show that subjectively, consumer-grade smartwatches are suitable to be used within clinical- and research-grade environments. The sensor acceptability study demonstrated high approval ratings from hospital patients and healthcare professionals for use of wearable motion sensors for continuous motion tracking. We found that in-patients were generally neutral towards or had no comment about the sensors in open responses; and strongly agreed with closed-ended statements that the sensors were simple to use, comfortable, unobtrusive and did not interfere with daily activities. At the same time, a small number of participants did raise worries with regards to discomfort, bulky sensor size, data privacy, and damage and loss. These concerns were similarly expressed across earlier wearable sensor usability studies [22]. For example, Tran, Riveros and Ravaud (2019) found that patients’ data privacy concerns included hacking of data and devices, spying on patients, and using and selling patient data without consent. Our findings highlighted that patients were considerably influenced by superficial characteristics related to sensor design and appearance, such as sensor colour schemes. This mirrors earlier findings, such as [20] which reported positive user opinions with regards to ‘colourful’, ‘beautiful’, ‘lightweight’ designs and ‘ease of use’ of evaluated smartwatches and [22] which recorded ‘lack of attractive features’ as the top concern for wearable devices. Acceptability results from healthcare professionals also revealed conflicting views regarding wearable sensor motion tracking across different medical specialties, such as the perceived opportunity of wearable movement sensors and artificial intelligence in healthcare applications. This highlights that different members of multidisciplinary teams have different experiences and expectations of technologies such as motion trackers, which need to be addressed when introducing such systems into clinical environments.

The device feasibility study had several strengths. The decision to adopt or use a new technology frequently involves a shared agreement between both the patients and healthcare professionals. Our study looked at both perspectives through the questionnaires. Our combined methods design, using open-ended and closed-ended questions, allowed broad insights into perceptions of the use of wearable sensors for continuous monitoring in a hospital setting. Comparable to earlier smartwatch usability studies [20], we used a Likert scale evaluation to enable us to gauge degrees of opinion. Furthermore, the findings captured a diverse range of views from a large sample of in-patients (*n* = 44) with wide-ranging demographics and multiple comorbidities. This offers an advantage over earlier studies such as [20] which only captured views from seven healthy subjects in a community setting; [21,22] which collected data from larger samples (*n* = 388 and *n* = 2058 respectively), but only recruited healthy subjects online who reported some or no experience using the wearable wristwatches.

The feasibility study was limited by not using a validated device usability questionnaire, such as System Usability Scale (SUS) [41], as we wanted to explore a broader set of questions (as subjects were not using the wearables as such but wearing them) while keeping questions brief to ensure completion and compliance rates. Unlike the study [20], we did not include the evaluation of satisfaction with the smartwatch user interface or battery as part of the study and prevented user on-screen interactions. Given that we wanted to evaluate the inertial sensor primarily for recording movement for healthcare professional in-patient monitoring, we did not choose to assess in-patient views of other smartwatch features and functionalities (e.g., networking and applications). In future research, we plan to develop and assess the acceptability of custom device movement feedback visualisations. Our smaller sample size of healthcare professionals (*n* = 15) may not be representative of the broader population and the sample was unbalanced for different medical specialties and experiences. Views on the usability of wearables were based upon recording durations (1.5–24 h) which is shorter than typical in-patient stays of several days to weeks. Our study also gained opinions only from those of patients in neurology and stroke wards, which may not be representative of other clinical settings and care homes. The study only collected views on Apple Watch wearables, which may not generalise to other wearable inertial devices. However, this allowed us to incorporate brand-related influences (e.g., brand loyalty and attitudes) that play a role in end-user adoption and to address a gap in the literature for perceptions related to the use of consumer devices for continuous monitoring in hospital settings [53]. We believe that there is, in principle, no technological barriers in allowing other smartwatch platforms with appropriate programmable interfaces and high-accuracy IMUs to be developed and look forward to this rapidly developing consumer electronics domain developing common interoperability standards for measuring, collecting and deploying healthcare data and applications.

In summary, our sensor acceptability study showed that (1) hospital patients wearing motion tracking smartwatches for 1.5–24 h are positive about their use; (2) healthcare professionals involved in clinical monitoring also embraced wearable IMU technology but concerns that need to be addressed are data privacy, compliance, sensor loss, specificity and discomfort.

## 5. Conclusions

These results suggest that for continuous long-term behavioural monitoring of in-patients, consumer smartwatches (such as Apple Watch) can offer reliable inertial tracking. Albeit more so for measures relying on angular velocity than linear acceleration for Apple smartwatches. The implication of this on a clinical application, for example, to measure the proportion of time lying in bed, as opposed to ambulating, or to estimate physical disability, needs to be ascertained by further studies. Our feasibility results provide reassurance that consumer smartwatch motion tracking is generally acceptable for patients and staff in hospitals, where we can now proceed with deploying these consumer-grade technologies to easily collect and monitor natural behaviour on a daily-basis from in-patient and care home residents. This may pave the way for improved care, patient safety, and novel data-driven solutions enabled by the availability of low-cost, high-accuracy natural behavioural data streams (e.g., [54,55]) that can be collected in a low-cost, accurate, continuous and socially distanced manner. The validation and end-user acceptance of these wearable sensors have important implications for the democratisation of healthcare, as the system can be used to cost-effectively improve patient monitoring, safety and care irrespective of the staff devoted to caring for any one patient [56]. Our results show that consumer-grade smartwatch use effectively provides researchers and healthcare technology developers with an accurate and acceptable platform enabling 24/7 watch over a patient.

## Figures and Tables

**Figure 1 sensors-20-07313-f001:**
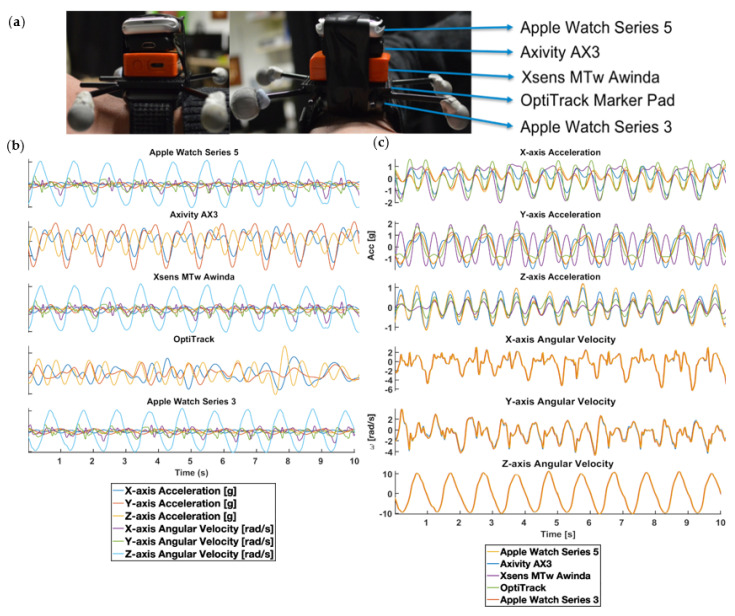
(**a**) depicts the participant wearing the sensor stack attached to the right wrist with the adjustable wrist strap. From top to bottom of the sensor stack, we mounted Apple Watches Series 5, Axivity AX3, Xsens MTw unit, OptiTrack retro-reflective marker pad with 4 retro-reflector unit (grey spheres), and Apple Watch Series 3 to one another vertically, aligning the centre of gravity, using Velcro sticky pads. (**b**) depicts 10-seconds sensors reading for each of the motion sensors. (**c**) shows 10-seconds sensors reading for each of the recording dimensions. The data depicted in (**b**,**c**) are sampled from one of the subject’s Composite Cross-Body Movement session and are both temporarily and spatially aligned.

**Figure 2 sensors-20-07313-f002:**
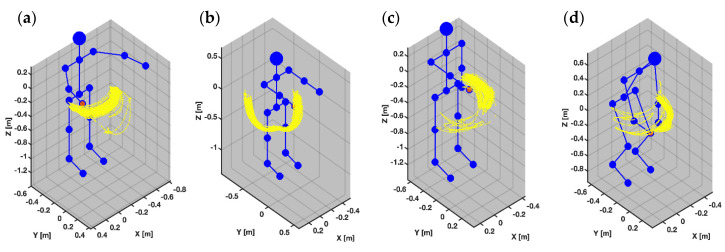
Depicts in blue the action component order for the participant performing the 4 movement tasks during the sensor validation protocol. Illustrated in yellow the OptiTrack reflective marker movement path associated with each movement task. Depicted in blue is a handcrafted skeleton used to establish a spatial reference. From top to bottom, the 4 movement tasks are: (**a**) Horizontal Arm Movement, (**b**) Vertical Arm Movement, (**c**) Rotational Arm Movement, and (**d**) Composite Cross-Body Movement.

**Figure 3 sensors-20-07313-f003:**
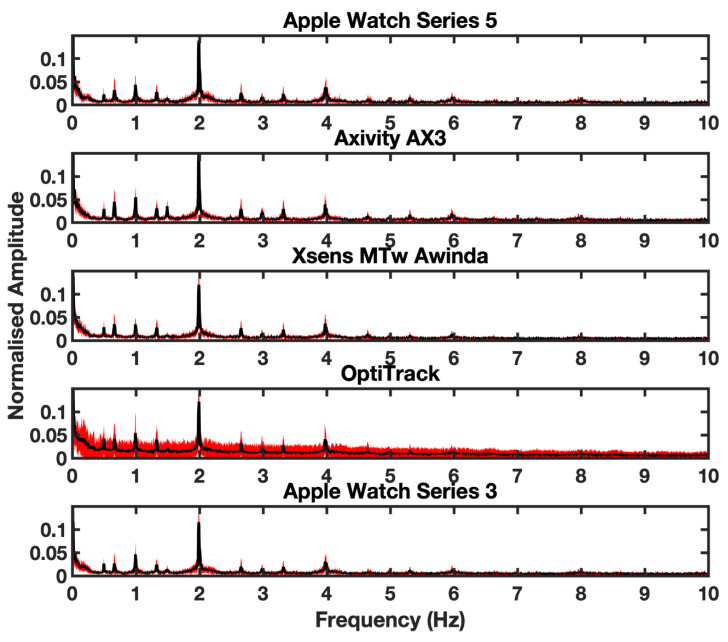
Depicts the power spectral density for the linear acceleration signal for each aggregated across all validation study participants. Shaded areas indicate 1 SD from the mean power, each row represents data from a different device. All evaluated device data and optical motion tracking were collected simultaneously.

**Figure 4 sensors-20-07313-f004:**
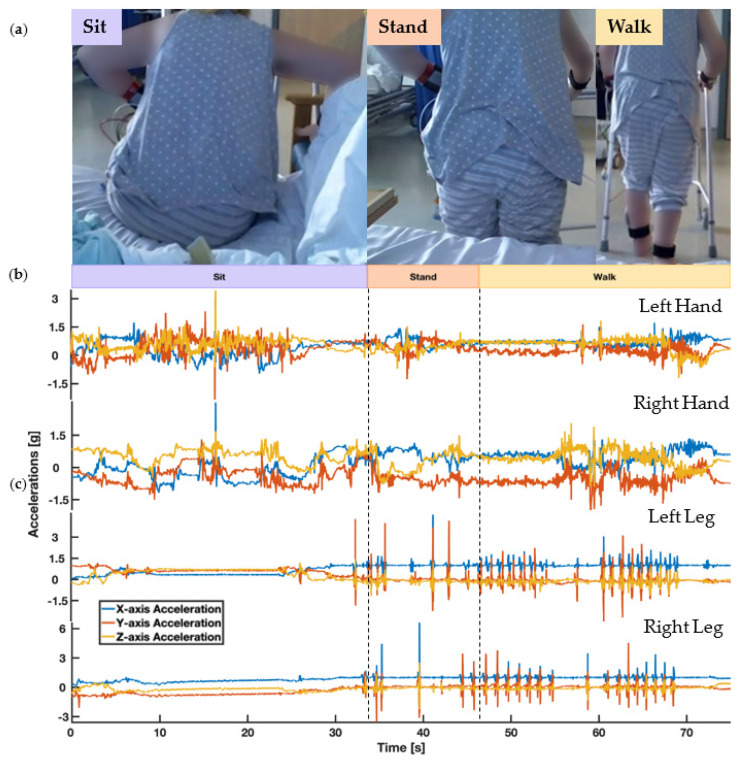
Depicts the participant’s behaviour while wearing the 4 wrist and ankle-worn sensors on the hospital ward and the associated sensors’ triaxial acceleration readings. (**a**) depicts the ground-truth video recording of the subject. (**b**) describes the associated behaviour labels. (**c**) shows the associated 3D linear acceleration signals of the 4 sensors.

**Figure 5 sensors-20-07313-f005:**
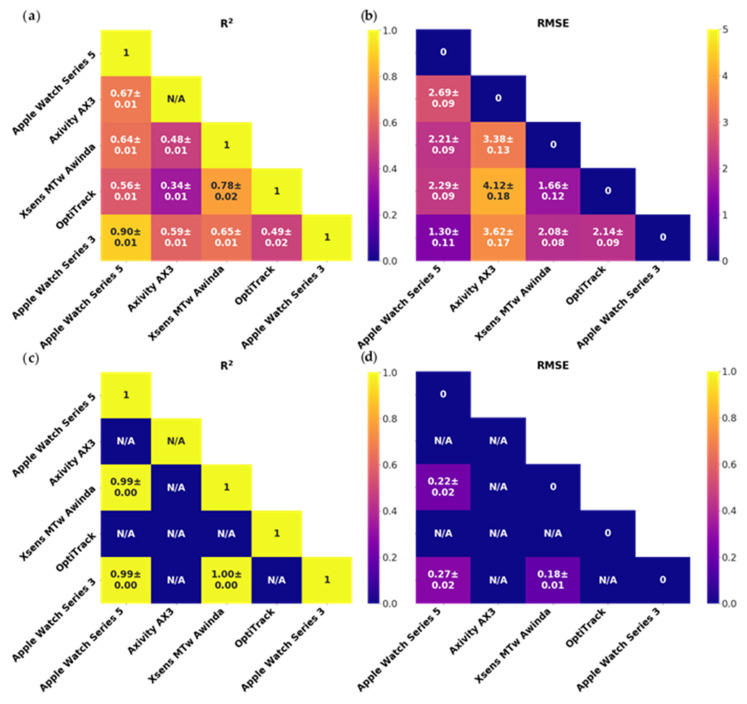
Triangle diagrams of the sensor validation study aggregated by pairwise signal measures (R^2^ for dynamic tracking of signals and RMSE metrics for offsets) between the selected motion sensors (Apple Watch Series 3 and 5, Xsens, Axivity, and Optitrack). Data is organised in (**a**) linear acceleration R^2^, (**b**) linear acceleration RMSE (m·s^−2^), (**c**) angular velocity R^2^, and (**d**) angular velocity RMSE (rad·s^−1^). Displayed R^2^ and RMSE values in the figure are rounded.

**Table 1 sensors-20-07313-t001:** Healthcare Professional and in-patient Questionnaire.

**In-Patient Closed Questions**	
icQ1.	The device was easy to put on and take off?
icQ2.	I would feel comfortable wearing the device even if it is visible to others?
icQ3.	I feel I could do most of my normal activities (except those involving water) wearing the device?
icQ4.	The device did not interfere with washing or going to the toilet?
icQ5.	I would find it easy to learn to use the device?
icQ6.	I did not experience any itchiness or skin irritations using the device?
icQ7.	I did not experience any discomfort wearing the device?
icQ8.	I did not feel anxious wearing the device?
icQ9.	I would be willing to wear the device continuously for long term use?
icQ10.	I did not find the appearance or design of the sensors obtrusive?
**Healthcare Professional Closed Questions**	
hcQ1.	The device was easy to put on and take off?
hcQ2.	I would find it easy to learn to use the device?
hcQ3.	Do you think that the increasing use of wearable tracking technology and Artificial Intelligence in healthcare is an opportunity?
hcQ4.	Do you think that the increasing use of wearable tracking technology in Artificial Intelligence in healthcare is a danger?
hcQ5.	If there were strong clinical evidence that the intervention would be equivalent or better than current neurological observations alone in a Neurology and Stroke setting, would you agree to use the new intervention in your own management of your patients?
**In-patient Open Questions**	
ioQ1.	What do you like about the device?
ioQ2.	What sort of characteristics and functions do you expect from the device?
ioQ3.	Is there anything you don’t like about the device?
**Healthcare Professional Open Questions**	
hoQ1.	What do you like about the device?
hoQ2.	What sort of characteristics and functions do you expect from the device?
hoQ3.	Is there anything you don’t like about the device?
hoQ4.	What do you think are the benefits and risks you perceive when using these new technologies?

Q =question, ic = in-patient closed; hc = healthcare professional closed, io = in-patient open; ho = healthcare professional open.

**Table 2 sensors-20-07313-t002:** Closed-ended Question Results for In-patients and Healthcare Professionals.

**In-Patient Questionnaires**	**Under 65**			**Over 65**		
	**Strongly Disagree**	**Somewhat Agree**	**Strongly Agree**	**Strongly Disagree**	**Somewhat Agree**	**Strongly Agree**
icQ1.	1	0	20	1	2	19
icQ2.	0	2	19	0	3	19
icQ3.	0	1	20	0	1	21
icQ4.	1	1	20	3	1	18
icQ5.	1	0	20	1	4	17
icQ6.	1	0	21	0	1	21
icQ7.	2	1	19	0	1	21
icQ8.	2	0	20	0	0	22
icQ9.	2	1	19	1	4	17
icQ10.	0	3	18	1	0	21
**Healthcare Professional Questionnaires**	**Strongly Disagree**		**Somewhat Agree**		**Strongly Agree**	
hcQ1.	2		0		13	
hcQ2.	2		0		13	
	No Opportunity			Great Opportunity		
hcQ3.	6			9		
	Dangerous			Safe		
hcQ4.	0			15		
	Would not use		Would only use if human-controlled		Would use and replace	
hcQ5.	0		6		9	

Table 2 describes (1) the frequency of responses for each descriptive category for in-patients (*n* = 44); (2) the frequency of responses for each descriptive category for healthcare professionals (*n* = 15). icQ1–10 and hcQ1–2 used a 1 to 7 rating scale: 1 to 2 (strongly disagree); 3 to 4 (somewhat agree); and 5 to 7 (strongly agree). hcQ3 used a 1 to 10 rating scale: 0 to 5 (no opportunity); and 6 to 10 (great opportunity). hcQ4 used a 1 to 10 rating scale: 0 to 5 (no danger); and 6 to 10 (danger). hcQ5 was collected with a −3 to +3 rating scale: −3 (would not use the intervention); −2 to 0 (would only use the intervention if controlled by a human caregiver); and 1 to 3 (would use the intervention and it could replace some interventions currently implemented by human caregivers).

**Table 3 sensors-20-07313-t003:** Themes of perceived likes, dislikes, expected functions and characteristics of technology from in-patient survey (ioQ1–3).

Themes	Details and Example Quotes
**Likes**	
Attractive appearance	‘Fine’, ‘good’, ‘stylish’, ‘beautiful’, ‘modern’, ‘simple’ design
Unobtrusive	‘unobtrusive’, ‘neutral’ or ‘unaware’ of the device. Sensor felt just like a ‘normal watch’, ‘non-invasive’
Ease of use	‘Easy to wear’, ‘simple to wear’
Visibility to others	Not concerned about the visibility of sensors to others as ‘appearance is fine’
Promising healthcare applications	‘Helpful for research’ and can improve healthcare
**Dislikes**	
Poor appearance	Would like a ‘colour scheme’
Discomfort	Skin ‘irritation’ from sensors
Interference with medical equipment	Cumbersome to wear with other ‘medical contraptions’
Poor straps	Would like ‘stretchy’ and ‘magnetic straps’, straps ‘hard to get on’
Large size	‘too big’
**Expected characteristics and functions**	
Well-designed straps	‘Better looking’ straps
Attractive appearance	‘beautiful design’, ‘brighter’ colour scheme
Time and heart rate functionalities	‘Heart rate’ and ‘time’ functionalities
Suitability for medical scans	‘Suitable’ to wear when going for ‘MRI scans’
Promising healthcare applications	‘Helpful for research’, ‘beneficial for other patients’
Smaller size	‘Smaller’ size

**Table 4 sensors-20-07313-t004:** Themes of perceived benefits, risks, likes, dislikes, expected functions and characteristic of technology from the healthcare professional survey (hoQ1–4).

Themes	Details and Example Quotes
**Benefits**	
Promising healthcare applications	‘Able to monitor movements to the development of new therapy’
Treatment personalisation	‘Tailored therapy’
Patient engagement	Way of engaging with patients in their own health
Patient tracking	‘Wearer could be tracked’ to know ‘where they are’
Unobtrusiveness	‘Gather information … in objective way & patients didn’t seem inconvenienced’
Ease of use	Ease of use and quick set-up
Convenience	‘convenient in the modern days of medicine’
**Risks**	
Data privacy risks	‘Ability for it to be shared with others that a patient did not consent to’
Sensor loss	‘It can be lost as it is easy to remove’
Discomfort	‘Not comfortable on skin and can contribute to skin wounds’
Specificity	‘Risk of false-positive results’
Compliance	Use …’ depends on patient compliance’
**Likes**	
Ease of use	‘Easy to wear and use’
Promising healthcare applications	System used for health monitoring
Lightweight	Lightweight
Unobtrusive	System is not obtrusive for patients or healthcare professionals
High portability	It’s ‘portable’ so it is possible to ‘monitor’ whilst the patient is ‘mobile’
**Dislikes**	
Large size	‘Too bulky’
Unsuitable for medical scans	Frustrated by having to ‘remove’ the sensors to accommodate ‘medical scans’
**Expected characteristics and functions**	
Alarm systems	Alerts to dangerous changes in ‘symptoms’ e.g., ‘GCS scores’ and area breaches within the ward
Integration vital measurements	Combine with other important clinical measurements e.g., ‘Temperature ‘and ‘peripheral capillary oxygen saturation’.
On-screen instructions	‘Instructions’ on how to use device.
Notifications	On-screen ‘notifications’ on device

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
