# Peer review of "Accuracy and Acceptability of Wearable Motion Tracking for Inpatient Monitoring Using Smartwatches"

_sensors, 2020, doi:10.3390/s20247313_

Round 1

Reviewer 1 Report

The current paper evaluates the accuracy and acceptability of consumer-grade wearable IMUs for the purpose of in-patient monitoring. Accuracy was measured by comparing the raw sensor data from consumer-grade IMUs to acceptable gold standards (optical tracking system and research-grade IMU) during movements of the upper extremity for a small sample of healthy participants (n = 12). Acceptability was measured by deploying a survey onto a convenience sample of in-patients (n=44) and healthcare professionals (n=15). Overall, I would like to commend the authors on a well-written manuscript. The authors topic/findings are impactful and will be of interest to the Sensors readership. To improve this manuscript, I have some minor revisions for the authors to consider (detailed below).

  1. Introduction (Page 2, Lines 48-52) – consider also describing the data obtained from a magnetometer, and how these data are often used to improve sensor fusion/drift.
  2. Introduction (Page 2, Lines 78-83) – I think it would be worthwhile to note whether there is any overlap in the MEMS IMU component suppliers for each commercial grade device listed here. In many cases these components in each commercial grade sensor may be supplied by a common third party (i.e. Bosch Sensortec, InvenSense, STMicroelectronics). Rather than comparing consumer smartwatch companies, it may be best to compare the suppliers of the MEMS IMU components themselves, as I would expect that the components from third party component suppliers would output similar raw-data, and may be present in multiple consumer-grade smartwatches.
  3. Introduction (Page 3, Lines 103-112) - Consider breaking up this sentence. Also, I’d suggest that the authors elaborate on the final statement. Specifically, in a fitness tracker context, it may be worth commenting on whether underestimation or overestimation errors of step-count data are preferred.
  4. Introduction (Page 3, Lines 140-142) – The justification of the sensors used for the current study (i.e. market share Q1 2020) stands out. If market share data were used to guide the selection of the devices, I would expect these data to be less current (i.e. from the study onset).
  5. Methods (Page 4, Lines 154-157) – Please comment on what scripting language was used to generate this custom software, and whether this software is being made publicly available (Open-Source).
  6. Methods (Page 4, Lines 164-167) – Please provide further information for how each sensor was affixed to the rigid body. To optimize accuracy between the sensors and gold-standard it is imperative that the sensors are not able to move relative to the rigid body.
  7. Methods (Page 4, Lines 167-169) – Two comments: (1) you cannot track the triaxial rotation of a single marker (only a rigid body composed of at least 3 markers); (2) Please correct the spelling of ‘Motiv’ to ‘Motive’ throughout.
  8. Methods (Data Processing) – given the small capture volume for the optical system used (and the relatively small number of markers used to define the rigid body), I am concerned about the prospect of missing data. The authors describe the use of Hampel filters to identify and interpolate outliers, were any additional interpolants used to gap-fill missing data in Motive? What was the typical size of a gap where interpolants were used? A poor interpolant could reduce the apparent accuracy of the wearable sensors relative to the gold-standard.
  9. Methods (Page 5, Lines 242-244) – Please describe the locations of each sensor/strap using additional anatomical detail.
  10. Results (Page 7, Lines 308-309) – Please include participant demographics for the validation study.
  11. Results (Page 7, Lines 338-349) – Consider presenting the survey responses for different age groups in your in-patient and healthcare groups. It may be possible that different age groups may have different preferences regarding the adoption of wearable tech into the clinic.
  12. Discussion (Page 8, Lines 378-383) – this section suggests that each sensor has some degree of on-board ‘black box’ processing. If this is the case, the current paper does not compare raw sensor data quality, but rather pre-processed data from each sensor (i.e. sensor fusion or filtering). Can the authors please confirm if the raw sensor data from each device were used in the sensor signal quality study?
  13. Discussion (Page 8, Lines 386-390) – I do not think that these statements are generalizable. These statements only consider the two smartwatches that are being compared. Future and past generations may not maintain the noted high level of intergenerational consistency.
  14. Figure 1 – Consider adding units and labels to the y-axes for (c). Further, please comment on if these data have been aligned spatially/temporally. In some cases, the XSens data look substantially different from all others (e.g. angular velocity). Finally, please comment on the movements being completed in (b) and (c) in the figure caption.
  15. Figure 3 – The authors only describe the tracking of a single rigid body placed on the wrist. Where are the remaining 3D post data coming from (depicted in (a))?
  16. Figure 4 – Please include a colourbar legend do that the reader can make sense of the heatmap being used.

Reviewer 2 Report

The authors address the problem of comparing the common consumer smartwatches and research-grade IMUs inertial accuracy relative to ground truth optical motion tracking.

Though the paper is mainly well structured and its content is relevant, some improvements could help to increase the readability of it. I would propose to make the following changes:

  1. Before the data collection section, a device description will be needed. To accomplish this, I suggest moving there the lines 288-297.
  2. In line 155 is said that some WachtOS app was developed, but there are no details about it. Please, consider to include some information about algorithm, libraries used, developing framework, etc. which could help the reader to have a more detailed view of the experimental set and about the influence of the software in the results.
  3. In section 2.1.4 is discussed the possible effect in the readings of the stack position of the devices. There is a mention of some plotting of the possible error but the reference of any figure is missing. What a surprise when I found this in Figure A1. Please, consider adding a reference to that figure in that paragraph.
  4. I would like to find all the device’s information ordered in the same way in the graphical and tabular representation of results. This could make more understandable and comparable the results. I suggest taking its order in the stack when the results of the different devices are shown in figures 1, 2 and 4.
